# Protein Source Determines the Effectiveness of High-Protein Diets in Improving Adipose Tissue Function and Insulin Resistance in *fa/fa* Zucker Rats

**DOI:** 10.3390/nu17203225

**Published:** 2025-10-14

**Authors:** Fadi H. J. Ramadan, Peter Zahradka, Carla G. Taylor

**Affiliations:** 1Department of Food and Human Nutritional Sciences, University of Manitoba, Winnipeg, MB R3T 2N2, Canada; fadi.ramadan@umanitoba.ca; 2Canadian Centre for Agri-Food Research in Health and Medicine, St. Boniface Albrechtsen Research Centre, Winnipeg, MB R2H 2A6, Canada; pzahradka@sbrc.ca; 3Department of Physiology and Pathophysiology, University of Manitoba, Winnipeg, MB R3E 0J9, Canada

**Keywords:** obesity, adipocytes, fat distribution, pancreatic islets, diabetes, insulin resistance, high-protein diet, animal protein, plant protein, *fa/fa* Zucker rats

## Abstract

Background/Objectives: Obesity and insulin resistance are increasing globally. Emerging evidence suggests that not only the quantity but also the source of dietary protein may improve metabolic health outcomes. This study evaluated the effects of high-protein diets (HPDs) containing animal or plant protein sources on obesity and obesity-related metabolic markers in a rodent model of genetic obesity. Methods: Obese male *fa/fa* Zucker rats were fed HPDs (35% of energy) containing protein from different sources (casein, egg white protein, soy + pea protein, mixture of egg white + soy + pea proteins) or a normal protein diet (15% of energy) containing casein over 8 weeks. Oral glucose tolerance, weight gain, fat depots, serum biochemistry, adipocyte and pancreatic islet size, and markers of adipose tissue lipolysis, insulin signaling, and immune cells were assessed. Results: Consumption of HPDs containing egg white protein, soy + pea, or their mixture resulted in smaller adipocytes compared to the casein diets, despite greater weight gain, elevated serum NEFA, and more total visceral fat in the HPD plant group. These HPD groups had reduced fasting insulin and no compensatory pancreatic islet enlargement. CD3 levels were elevated in adipose tissue without changes in F4/80, and no differences were observed in ATGL, HSL, Akt or AS160. Conclusions: The source of dietary protein in HPDs significantly influences metabolic outcomes in obese rats, impacting adipocyte and pancreatic islet size, insulinemia, and immune cell markers in adipose tissue. These findings support the potential of employing targeted dietary protein interventions for managing obesity-related metabolic disorders.

## 1. Introduction

Obesity and type 2 diabetes mellitus are widespread and complex metabolic disorders characterized by insulin resistance, persistent low-grade inflammation, and adipose tissue dysfunction [1,2,3]. Adipose tissue plays a pivotal role in these conditions, not only serving as a site for energy storage but also acting as an active endocrine organ that regulates systemic metabolic homeostasis [4,5].

A critical yet often underappreciated factor in shaping metabolic health is dietary composition, particularly the amount and source of protein [6]. High-protein diets (HPDs) with ~20–30% of energy from protein are effective for attaining weight loss in individuals with obesity and for improving glycemic control in persons with type 2 diabetes [6]. Given initiatives to increase consumption of plant-based proteins [7], there has been interest in whether the source of proteins included in HPDs alters metabolic outcomes. Recent human studies indicate that HPDs containing primarily plant-based or animal-based sources of protein have similar effects for improving glycemia and cardiometabolic parameters in the context of hypocaloric diets and weight loss for individuals with obesity and/or diabetes [8,9,10]. On the other hand, we have shown that an HPD containing a mixture of animal- and plant-based protein sources is more effective for reducing fasting insulin and insulin resistance than an HPD containing soy protein, despite no differences in body weight or adiposity [11]. Although animal-based and plant-based proteins differ in their amino acid composition, digestibility, stimulation of gastrointestinal hormones, and post-absorptive effects [6], their downstream impacts on metabolic pathways have received little attention.

Adipocyte hypertrophy, a hallmark of obesity, is more strongly associated with insulin resistance than total fat mass or body mass index [12]. Enlarged adipocytes exhibit increased secretion of pro-inflammatory cytokines, leading to systemic insulin resistance and recruitment of immune cells, particularly macrophages, into adipose tissue [3]. Conversely, smaller adipocytes are linked to improved insulin signaling and metabolic flexibility [12,13]. A previous study has shown that an HPD containing milk protein reduces adipocyte cell size in Wistar rats compared to normal protein diet [13], however, that study did not investigate parameters associated with insulin action or inflammation as the model was healthy rats.

Adiponectin, a hormone secreted by adipocytes, plays a protective role in metabolic regulation. Higher circulating adiponectin levels are associated with increased insulin sensitivity and reduced hepatic fat accumulation in individuals with obesity and type 2 diabetes, whereas low adiponectin levels are indicative of metabolic dysfunction across diverse populations [14,15].

Recent studies have also underscored the immunometabolic role of adipose-resident T cells. An increase in anti-inflammatory regulatory T cells (Tregs) has been shown to reduce adipose tissue inflammation and improve insulin sensitivity [16]. These findings highlight the potential of immune cell modulation in adipose tissue as a target for therapeutic intervention in metabolic diseases. However, research exploring how HPDs derived from different protein sources affect immune cells present in adipose tissue is lacking.

Different protein sources can modulate features of metabolic dysfunction such as adipocyte remodeling and pancreatic stress [11], but our understanding of how protein from different sources affects metabolism is limited. This knowledge gap is hampering our ability to optimize dietary interventions utilizing protein. The objective of this study was to evaluate the metabolic consequences of HPDs containing animal or plant protein sources in a genetic rat model of obesity, insulin resistance and metabolic dysfunction. Specifically, we assessed weight gain, adiposity, adipocyte size, oral glucose tolerance, pancreatic islet size, serum biochemistry and markers of adipose tissue lipolysis, insulin signaling, and immune cells. By comparing groups fed distinct protein diets, we sought to determine how different dietary protein sources modulate obesity-related outcomes, insulin resistance, and pancreatic compensation, with a particular focus on adipose tissue remodeling and its impact on systemic metabolism.

## 2. Materials and Methods

### 2.1. Animals, Diets and In Vivo Assessments

Six-week-old male *fa/fa* Zucker (Obese) rats (Charles River Laboratories, Saint-Constant, PQ, Canada; n = 48, strain code 185, Crl:ZUC-*Lepr^fa^*) were housed individually (in flat-bottomed cages with woodchip bedding and cardboard tubes for enrichment) under constant temperature (25 ± 1 °C) with a 12 h dark/light cycle. After a minimum 1-week acclimation period, the rats were randomly assigned to 6 groups (n = 8/group): the Baseline group, one of 4 high-protein diet (35% of energy from protein) groups for 8 weeks where the diets contained casein (HPcasein), egg white protein (HPegg white), plant protein composed of soy protein + pea protein (1:1; HPplant), or a mixture of egg white protein + soy protein + pea protein (2:1:1; HPmix), or a normal-protein casein group (NPcasein; 15% of energy from casein as per the American Institute of Nutrition-93G diet [17]) for 8 weeks. The diet formulations are shown in Table 1; protein replaced carbohydrate in the HPDs, and the fat content was constant among the diets. The rats had free access to the diets and water, and they were weighed weekly. Personnel feeding the diets were aware of the group allocations, whereas blinding was possible during the outcome assessments when rats were denoted by a 3-digit number. Whole body fat and lean composition were measured with an EchoMRI-700^TM^ (EchoMRI LLC, Houston, TX, USA), and oral glucose tolerance testing (OGTT) was conducted at week 0 for the Baseline group and during week 8 for the other groups. Rats were fasted for 5 h, and saphenous blood was collected before an oral glucose challenge (1 g glucose/kg body weight) and post-challenge at 15, 30, 60, and 120 min, and then centrifuged at 3400× *g* for 10 min for preparation of serum that was stored at −80 °C. The protocol was approved by the University of Manitoba Animal Care Committee (20-071). Effects of these diets on hepatic-related parameters (triglycerides, lipid droplet size and distribution, fibrosis, and proteins involved in lipid metabolism) in these animals are reported elsewhere [18].

### 2.2. Tissue Collection

At the end of the study, rats were deeply anesthetized with pentobarbital and euthanized by cervical decapitation. Trunk blood was collected and centrifuged at 3400× *g* for 10 min to obtain serum, which was stored at −80 °C for metabolic marker analysis. Visceral adipose depots (epididymal, peri-renal, mesenteric) and pancreas were removed, rinsed in cold phosphate-buffered saline (PBS; 137 mM NaCl, 2.7 mM KCl, 10.1 mM Na_2_PO_4_, and 1.8 mM KH_2_PO_4_, pH 7.4), and weighed. A portion of epididymal adipose tissue was embedded in OCT gel (Tissue-Tek, Torrance, CA, USA) and frozen in a dry ice–ethanol bath. Subcutaneous fat mass was calculated by subtracting visceral fat from the total fat mass obtained from body composition analysis. A portion of the pancreas was fixed in 10% phosphate-buffered formalin followed by paraffin embedding for histological examination. The remaining tissues were snap-frozen in liquid nitrogen and stored at −80 °C for further analysis.

### 2.3. Serum Biochemistry

Serum from the OGTT was analysed for glucose and insulin by colorimetric assay (Megazyme, Bray, Ireland) or U-PLEX immunoassay (MesoScale Discovery, Rockville, MD, USA), respectively. The fasting serum samples from the OGTT were also analyzed for non-esterified fatty acid (NEFA) by an enzyme colorimetric LabAssay NEFA kit (Wako-Pure Chemical, Richmond, VA, USA). The NEFA assay was performed according to the manufacturer’s instructions, and the results were analyzed with a FLUOstar Omega microplate reader (BMG Labtech Inc., Cary, NC, USA) at a wavelength of 550 nm. Non-fasting serum obtained at termination was analyzed for leptin and adiponectin by colorimetric U-PLEX immunoassay kits (Meso Scale Discovery, Rockville, MD, USA).

### 2.4. Histological Examination of Pancreatic and Adipose Tissues

Paraffin-embedded pancreatic sections (5 μm, every 10th section) were prepared and stained with hematoxylin and eosin (H&E) (ScyTek, Logan, UT, USA). Briefly, the sections were dewaxed in xylene followed by rehydration in descending ethanol series through water before they were incubated in H&E and dehydrated in ascending ethanol series, cleared in xylene, and mounted using Permount. Images were taken at 10× magnification with a Zeiss Axioskop-2 mot plus fluorescence microscope (Carl Zeiss Canada Ltd., Toronto, ON, Canada). Images of OCT-embedded epididymal adipose sections (10 μm) were captured at 20× with simple light inverted CX41 Olympus microscope (Olympus Canada Inc, Richmond Hill, ON, Canada). Pancreatic islet and adipose cell size were quantified using ImageJ software (version 1.53, National Institutes of Health, Bethesda, MD, USA) [19].

### 2.5. Western Blot Analysis

Frozen epididymal adipose tissue was incubated in chilled lysis buffer [50 mM Tris-HCl pH 7.4, 150 mM NaCl, 0.1% sodium dodecyl sulfate (SDS), 1% Triton X-100, 1 mM EDTA, 0.5% sodium deoxycholate, and protease-phosphatase inhibitor cocktail (1:200, MilliporeSigma, Oakville, ON, Canada)] at a 1:2 (*w*/*v*) ratio on ice for 15 min with continuous shaking. Then, the samples were sonicated two times for 7 s each and cleared by centrifugation at 18,000× *g* for 15 min at 4 °C. The protein concentration was determined with the Pierce BCA protein assay (ThermoScientific, Mississauga, ON, Canada). The lysates (20 μg protein) were heated at 100 °C for 5 min, and subjected to 10% or 12% sodium dodecyl sulfate polyacrylamide gel electrophoresis (SDS-PAGE) using a Triple Wide Mini electrophoresis and blotting system (C.B.S. Scientific, San Diego, CA, USA) to have all samples on the same gel. The protein was then transferred to nitrocellulose membranes, which were blocked in 5% bovine serum albumin (BSA) for 1.5 h at room temperature, and then incubated with primary antibodies diluted in blocking solution overnight at 4 °C. The following primary antibodies (Cell Signaling Technology, Danvers, MA, USA) were used: anti-ATGL, anti-HSL, anti-phospho-HSL (S563), anti-phospho-HSL (S565), anti-Akt, anti-phospho-Akt (Thr308), anti-phospho-AS160 (Thr642), anti-CD3, and anti-F4/80, all at 1:1000 dilution, and anti-actin at 1:2000 dilution. On the second day, appropriate horseradish peroxidase-conjugated secondary antibodies were used at a dilution of 1:1000 (anti-mouse and anti-rabbit IgGs; Cell Signaling). Chemiluminescence generated by applying the WesternBright ECL HRP substrate (Advansta, San Jose, CA, USA) was measured using a ChemiDoc MP high-resolution imaging system (Bio Rad, Mississauga, ON, Canada). Pixel volumes of the bands were determined using Image Lab software (version 6.1, Bio Rad). For membrane stripping and re-probing, the membranes were washed in a stripping buffer containing 0.2 M glycine and 0.1% SDS (pH 2.2) for 30 min at room temperature. After washing and blocking, the membranes were re-probed.

### 2.6. Statistical Analysis

Time course data (e.g., OGTT) and end-point data were analyzed using repeated-measures analysis of variance (ANOVA) and one-way ANOVA, respectively, followed by Duncan’s multiple range test using Statistical Analysis Software (SAS; Version 9.4, SAS Institute, NC, USA). Data were log transformed to achieve normality when necessary. Data that did not follow a normal distribution or were not homogenous were analyzed by Kruskal–Wallis, followed by Least Significant Difference (LSD) test. Outliers (greater or less than the overall mean ± 2.5 standard deviations) were removed from the dataset before analysis. Differences were considered statistically significant at *p* < 0.05.

## 3. Results

### 3.1. Weight Gain and Adipose Tissue Distribution

The overall mean body weight of the *fa/fa* rats was 172 ± 3 g (mean ± SEM) when randomized to the Baseline or experimental HPD groups. At the end of the 8-week study period, the *fa/fa* rats fed plant-based HPDs (HPplant and HPmix) had greater weight gain (Figure 1A) compared to those on either NPcasein or HPcasein diets. Weight gain of the HPegg white group was similar to *fa/fa* rats fed the casein diets. Interestingly, when expressed relative to body weight, epididymal fat mass was lower in the HPcasein group compared to the other groups (Figure 1B), and mesenteric fat in the HPmix group was reduced relative to both casein groups as well as in the HPegg white group compared to the NPcasein group (Figure 1C). Peri-renal fat was greater in the HPplant group compared to the other HPD groups (Figure 1D). Visceral fat was elevated in the NPcasein and HPplant groups compared to the HPcasein group (Figure 1E), while subcutaneous fat was lower in rats fed diets containing egg white protein (HPegg white and HPmix) compared to the HPcasein group (Figure 1F).

### 3.2. Adipose Tissue Morphology and Immune Cell Markers

Adipose tissue, as an endocrine and storage tissue, influences metabolic outcomes [4,5,20]. Given the critical roles of adipocyte morphology and infiltrating immune cells in regulating this function, we examined markers of adipocyte functionality. Analysis of adipocyte size showed more enlarged adipocytes in the casein diet groups than the other groups, while the HPmix group retained a cell size comparable to the Baseline group (Figure 2A,B). Hypertrophic adipocytes are associated with adipose tissue dysfunction [21]. The smaller adipocytes of HPegg white, HPmix, and HPplant groups suggest a metabolically healthier adipose phenotype despite the greater overall weight gain. Serum leptin levels were unchanged among experimental groups (Figure 2C). In contrast, serum adiponectin, an insulin-sensitizing adipokine often inversely correlated with adipose inflammation and insulin resistance [14,15], showed a modest elevation in the HPegg white, HPmix, and HPplant groups compared to the NPcasein group, while only the HPplant group showed higher levels than the HPcasein group (Figure 2D). A significant effect of plant protein consumption on serum adiponectin is indicated by these results. Western blot analysis of adipose tissue revealed a notable increase in CD3 levels in the HPegg white, HPmix, and HPplant groups relative to both casein groups (Figure 2E,G), while F4/80 levels remained unchanged among the experimental groups (Figure 2F,G). The selective elevation of CD3, a pan-T cell marker, without corresponding changes in the macrophage marker F4/80, suggests a shift in the T cell populations rather than general immune cell infiltration.

### 3.3. Adipose Tissue Lipolysis Markers

Lipid metabolism in adipose tissue primarily involves triglyceride storage and mobilization of NEFA, with the direction of flow (uptake from circulation vs. release into circulation) dependent predominately on the feeding/fasting state [22]. Fasting serum NEFA levels were elevated in the HPplant group compared to both casein groups and were higher in the HPegg white and HPmix groups compared to the NPcasein group (Figure 3A). To further assess lipid mobilization, Western blotting was used to examine lipolysis and insulin signaling in adipose tissue. No differences were found in ATGL or HSL (Figure 3B–F) across HPD groups. The HPegg white group had elevated p-HSL-S563 (stimulates activity) compared to HPcasein, and the HPmix group had higher p-HSL-S565 (inhibits activity) compared to HPcasein, but otherwise phosphorylated HSL (activated/inhibited forms) was not different among the HPD groups. Akt, p-Akt, and p-AS160, key components of the insulin signaling cascade [23], remained unchanged among HPD groups (Figure 3B,G–I), indicating preserved basal insulin signaling in adipose tissue. These findings suggest that the elevated levels of circulating NEFA in the HPegg white, HPmix, and HPplant groups are not driven by enhanced adipose lipolysis.

### 3.4. Glucose and Insulin Homeostasis

Since hypertrophic adipocytes are associated with impaired insulin sensitivity [24], glucose and insulin were evaluated during fasting and in response to an oral glucose challenge. Fasting serum glucose levels remained unchanged across all groups (Figure 4A). However, *fa/fa* rats consuming the casein diets exhibited a 3-fold increase in the fasting insulin concentrations relative to the other experimental groups (Figure 4B), indicative of insulin resistance and a compensatory mechanism where the pancreas secretes more insulin to maintain normoglycemia [24]. All experimental groups had impaired oral glucose tolerance compared to Baseline as demonstrated by the elevated serum glucose concentrations at 30 and 60 min (Figure 4C). The NPcasein and HPcasein groups were not different from the Baseline group at 15 min or 120 min. The HPegg white group had lower serum glucose at 30 min compared to the HPmix and HPplant groups. The serum glucose of all groups returned to Baseline levels at 120 min except for the HPmix and HPplant groups. There were distinct patterns in the serum insulin concentrations during the oral glucose tolerance test (Figure 4D). The Baseline group had the lowest levels and the NPcasein group had the highest levels (5-fold higher than Baseline) at all time points. Al-though the HPegg white and HPmix groups had serum insulin 2-fold higher than the Baseline group, this was not statistically different. At 0, 15 and 30 min, the HPcasein and HPplant groups had an intermediate level of serum insulin not different from any of the other groups. At 60 and 120 min, the serum insulin of the HPcasein and HPplant groups remained elevated compared to the Baseline group. The HPegg white group had reduced AUC-glucose compared to HPmix and HPplant groups (Figure 4E), while AUC-insulin was ~50% lower in HPegg white, HPmix and HPplant groups compared to NPcasein (Figure 4F).

### 3.5. Pancreatic Morphology

Serum insulin levels are mainly regulated by pancreatic β-cells, and hyperinsulinemia is a sign of pancreatic dysfunction or overfunctioning [24]. For this reason, we examined pancreatic islet morphology (Figure 5A). There were no significant differences in overall pancreatic weight relative to body weight among the experimental groups (Figure 5B). However, histological analysis revealed a marked increase in pancreatic islet size in casein-fed rats compared to the other groups (Figure 5A,C). This expansion of islet area is a hallmark of β-cell hypertrophy, commonly observed as a compensatory response to insulin resistance [24]. Also, the islet size of HPmix rats was comparable to that of Baseline rats and, along with their smaller adipocyte size and normal fasting insulin levels, suggests less β-cell stress or demand.

## 4. Discussion

This study investigated how different dietary proteins from plant and animal sources, either alone or in combination in HPDs, influence metabolic health in an obese rat model. Our novel findings demonstrate that the type of dietary protein in HPDs is capable of modulating a variety of obesity-related outcomes, including adipose tissue function, glucose–insulin responses, and pancreatic islet size, independent of overall fat mass and without weight loss. Of note, the advantages of egg white protein or plant protein (soy + pea) when used individually in an HPD were maintained when they were provided in combination. Interestingly, two animal-based proteins, casein and egg white protein, elicited opposite outcomes with respect to adipose function and insulin action, indicating their impact on metabolic pathways is markedly different.

The *fa/fa* rats fed HPDs containing egg white protein, plant protein, or a mixture of these protein sources had smaller adipocytes in epididymal fat, comparable to Baseline levels, suggesting a protective effect against adipocyte hypertrophy. Additionally, these HPDs improved insulin sensitivity compared to both casein diets as supported by the presence of lower fasting insulin and smaller pancreatic islets. The postprandial insulin response and AUC-insulin were also lower compared to the NPcasein group. These outcomes are consistent with previous findings that smaller adipocytes are metabolically favorable and correlate better with insulin sensitivity than total body weight and fat mass [12,21,25]. In fact, the improvements in adipose function and insulin sensitivity in these groups occurred despite greater weight gain by the HPmix and HPplant groups, with only minor differences in fat depot size. In contrast, the casein-fed groups, regardless of the protein level, exhibited adipocyte hypertrophy, elevated fasting and postprandial insulin, and enlarged pancreatic islets, suggesting pancreatic compensation for insulin resistance. Furthermore, the unchanged fasting serum glucose across groups, along with increased pancreatic islet size in both casein-fed groups, reinforces the compensatory insulin hypersecretion characteristic of early-stage insulin resistance [26,27]. Interestingly, the HPegg white, HPmix, and HPplant groups maintained lower fasting serum insulin levels and had better glucose–insulin homeostasis in response to an oral glucose challenge, despite elevated fasting serum NEFA compared to NPcasein. Although this might suggest increased lipid mobilization without concurrent hepatic or peripheral insulin resistance, markers of lipolysis (ATGL and HSL) and key components of the insulin signaling cascade (Akt and AS160) were unchanged in adipose tissue. These findings suggest that dietary protein composition may exert effects through immune or endocrine modulation rather than direct changes in the biochemical pathways regulating lipid metabolism in adipose tissue. Additionally, inter-organ metabolism needs to be considered, and the *fa/fa* rats fed HPDs containing egg white protein, plant protein, or a mixture of these protein sources also had ~50% reduction in hepatic fat accumulation [18], which would also contribute to improvements in insulin-related parameters.

Immune cell infiltration is another aspect affecting adipose dysfunction. The HPegg white, HPmix, and HPplant groups exhibited increased levels of CD3, a T cell marker, in epididymal adipose tissue, with no changes in F4/80, a macrophage marker. Given that CD3 is indicative of all T cells, this elevation may indicate an increase in Tregs, which are known to exert anti-inflammatory effects and improve insulin sensitivity [20,28]. This interpretation is supported by the reduced fasting insulin, AUC-insulin and pancreatic islet size for these groups. The lack of change in macrophage infiltration, as reflected by F4/80, suggests the adipose tissue becomes enriched with Tregs rather than generalized adipose inflammation, which would be indicative of a selective immunomodulatory effect by the dietary protein sources. This premise is in agreement with the reduced adipocyte size and higher serum adiponectin levels observed in HPegg white, HPmix, and HPplant groups compared to NPcasein. Future investigations of the potential immunomodulatory effects of HPDs need to employ flow cytometric analyses to discern the response of specific immune cell sub-types, including Tregs, in adipose tissue.

Although there were some differences in weight gain and body fat distribution among the experimental groups, these did not translate into significant impacts on adipose function or glycemic responses. For example, the HPD groups consuming plant protein (HPplant, HPmix) had greater weight gain (~15%) than the other HPD groups, more visceral fat (epididymal and peri-renal) than the HPcasein group, a slightly larger adipocyte size than the HPmix and HPegg white groups, and higher AUC-glucose than the HPegg white group. On the other hand, the adipose CD3 and F4/80 profile, fasting insulin, AUC-insulin, and pancreatic islet size of the plant protein groups were similar to the HPegg white group.

The comparison of different dietary protein sources is a key element of this study. We had previously reported that HPDs containing a mixture of animal and plant proteins (egg white, milk, soy, wheat) or soy protein were effective for modulating insulin resistance and hepatic steatosis independent of weight loss in older *fa/fa* rats with established obesity [11]. The current study extends these findings to HPDs containing egg white protein alone or in combination with soy + pea protein or these plant proteins alone, and shows that metabolic health can be positively altered during the growth stage in obese rats with protein intakes within the upper limit of the Acceptable Macronutrient Distribution Range (35% energy from protein) [29]. Studies in mice fed high-fat diets (60% energy from fat) have shown some benefits of whey protein compared to casein [30] or soy protein [31] on adipose-related parameters and insulin resistance in the context of lower body weights. Another strength of this study lies in its integrative approach, combining histological, molecular, and metabolic assessments to elucidate how distinct dietary protein sources modulate adipose tissue function and insulin sensitivity in obesity. However, the use of male rats limits the generalizability of findings to both sexes, given the known influence of sex on metabolic and immunological responses. Although changes in T cell and macrophage markers were observed alongside shifts in adipocyte morphology, the underlying molecular mechanisms remain incompletely defined. Other studies investigating dietary protein sources and obesity have been examining the potential roles of adipose tissue browning, lipid metabolism, and host-gut microbiota interactions [30,31,32], which also could contribute to the metabolic effects we have observed.

Limitations of the present study include not comparing male and female rats, and not recording energy and water intake. Hormonal effects and differences in body composition could contribute to sex differences in protein metabolism [33]. To date, sex differences have not been evaluated in pre-clinical [11,30,31,32] or human studies [8,9,10] investigating protein sources in HPDs, insulin resistance, and adipose-related parameters, and this needs to be addressed in future studies. In the present study, we do not know if the greater weight gain and final body weight of the HPD groups consuming plant-derived protein is due to increased feed intake, altered energy metabolism, or other factors. In our previous study, there were no differences in feed intake among *fa/fa* groups consuming HPDs containing casein, soy, or mixed protein, or with the control *fa/fa* group fed a normal-protein diet containing soy protein [11]. Interestingly, mice fed high-fat diets from a young age have increased feed intake but less body weight gain when consuming whey protein versus casein [32]. Thus, measurements of feed intake and energy expenditure need to be included in future HPD studies investigating dietary protein sources and their effects on body weight and metabolism.

## 5. Conclusions

In conclusion, our study demonstrates that HPDs containing certain protein sources (i.e., egg white protein and/or soy + pea protein) preserve adipocyte function and insulin sensitivity by preventing adipocyte hypertrophy and supporting anti-inflammatory immune profiles known to positively influence long-term metabolic health. Understanding how different protein sources modulate key features of metabolic dysfunction such as adipocyte remodeling, fatty liver disease, and pancreatic stress is essential for optimizing dietary interventions. Elucidating the mechanisms by which specific protein sources modulate metabolic pathways and exploring their potential in dietary interventions for metabolic disorders would be an appropriate focus for future research aimed at determining which dietary formulations are suitable for therapeutic utilization.

## Figures and Tables

**Figure 1 nutrients-17-03225-f001:**
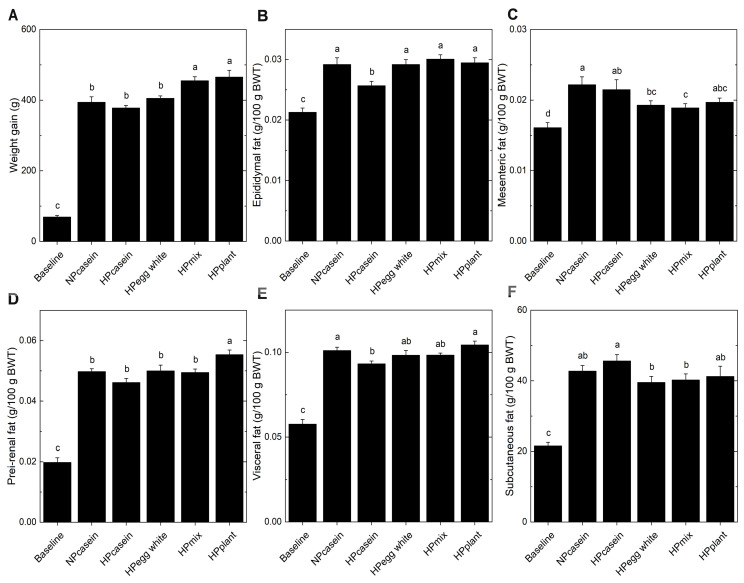
Weight gain and fat depots. (**A**) Total weight gain over 8 weeks, and (**B**) epididymal fat, (**C**) mesenteric fat, (**D**) peri-renal fat, (**E**) visceral fat, and (**F**) subcutaneous fat relative to body weight at end of study. Data are means ± SEM (n = 8 per group, except n = 7 for the Baseline group in (**B**) and (**E**)). Columns with different letters are significantly different (*p* < 0.05) from each other. Abbreviations: BWT, body weight; HPcasein, high-protein casein diet; HPegg white, high-protein egg white diet; HPmix, high-protein mix diet; HPplant, high-protein plant diet; NPcasein, normal-protein casein diet.

**Figure 2 nutrients-17-03225-f002:**
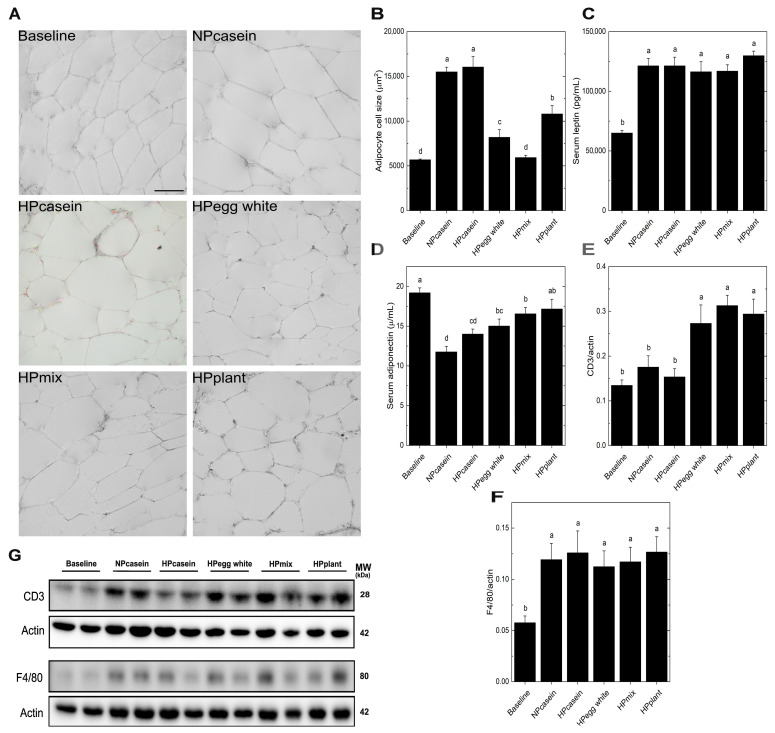
Adipose morphology, adipokines and immune cell markers. (**A**) Representative images of epididymal fat cells, (**B**) adipocyte cell size, (**C**) serum leptin, (**D**) serum adiponectin, and (**E**,**F**) densitometric quantification after normalization to actin and (**G**) representative Western blots for CD3 (T-cell marker) and F4/80 (macrophage marker) in epididymal adipose tissue. The scale bar shown in (**A**) for the Baseline group represents 20 μm and applies to all panels. Data are means ± SEM (n = 3 rats per group for (**B**) and n = 7–8 per group for (**C**–**F**)). Columns with different letters are significantly different (*p* < 0.05) from each other. Abbreviations: CD3, Cluster of Differentiation 3; MW, molecular weight; kDa, kilodalton.

**Figure 3 nutrients-17-03225-f003:**
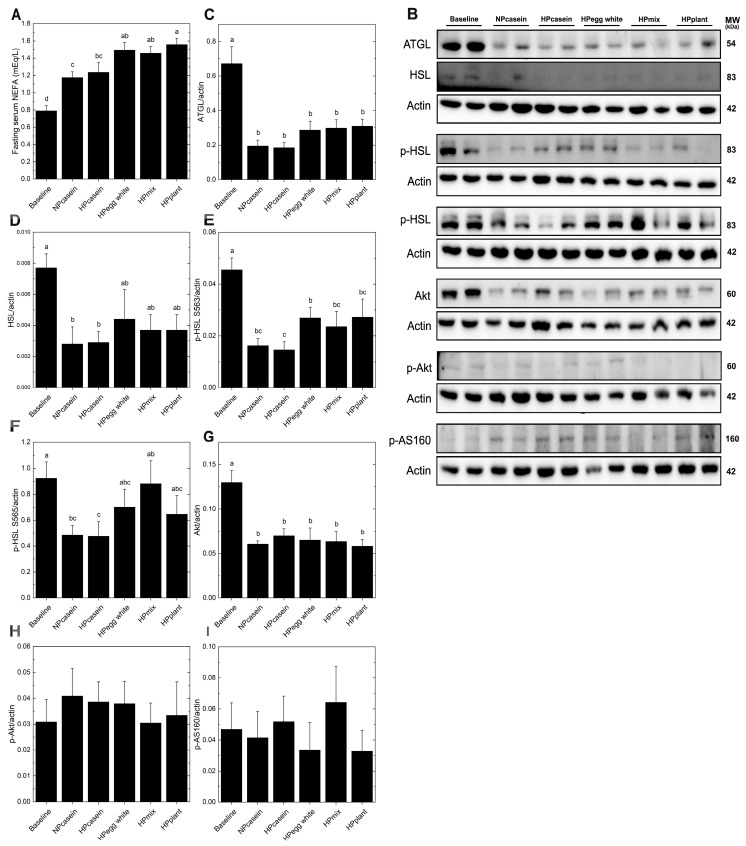
Adipose lipolysis markers. (**A**) Fasting serum NEFA, (**B**) representative Western blots and relative band intensity after normalization to actin for (**C**) ATGL, (**D**) HSL, (**E**) p-HSL S563, (**F**) p-HSL S565, (**G**) Akt, (**H**) p-AKT, and (**I**) p-AS160. Data are means ± SEM (n = 5–8 rats per group); columns with different letters are significantly different (*p* < 0.05) from each other. Abbreviations: ATGL, adipose triglyceride lipase; HSL, hormone-sensitive lipase.

**Figure 4 nutrients-17-03225-f004:**
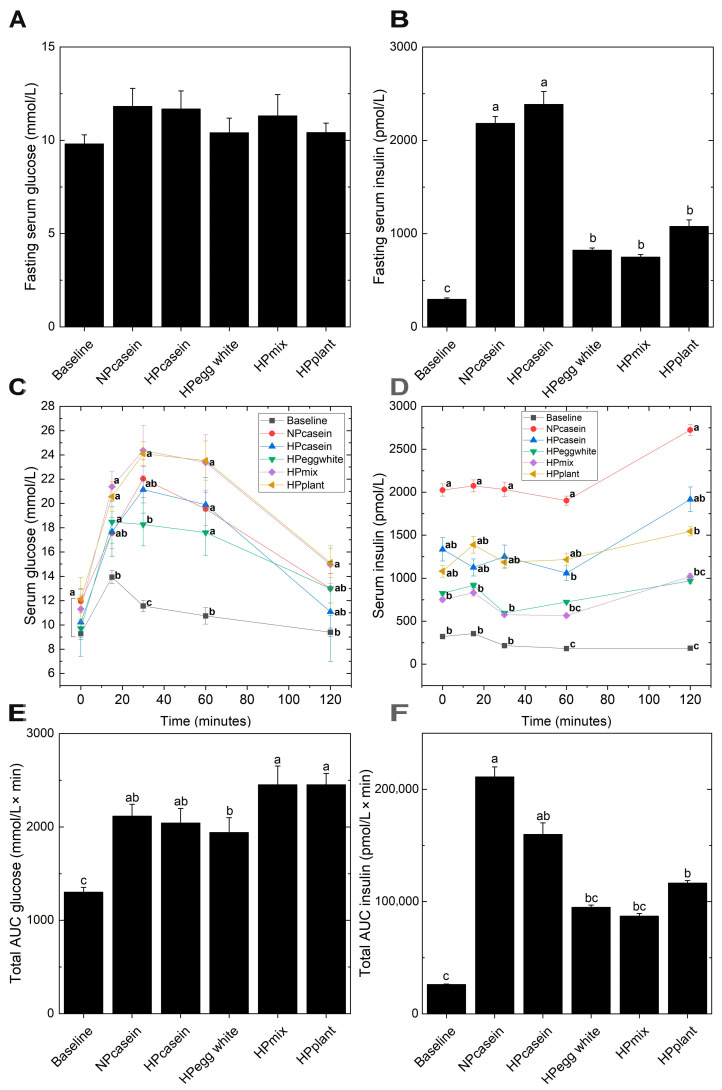
Glucose and insulin homeostasis. (**A**,**B**) Fasting serum glucose and insulin, (**C**,**D**) serum glucose and insulin and (**E**,**F**) AUC for glucose and insulin during an oral glucose tolerance test. Data are presented as means ± SEM (n = 7–8 per group for (**A**,**B**); n = 3–8 for (**C**–**F**)). Columns in (**A**,**B**,**E**,**F**) with different letters are significantly different (*p* < 0.05); in (**C**,**D**), different letters at the same time point indicate significant differences (*p* < 0.05). Abbreviations: AUC, area under the curve.

**Figure 5 nutrients-17-03225-f005:**
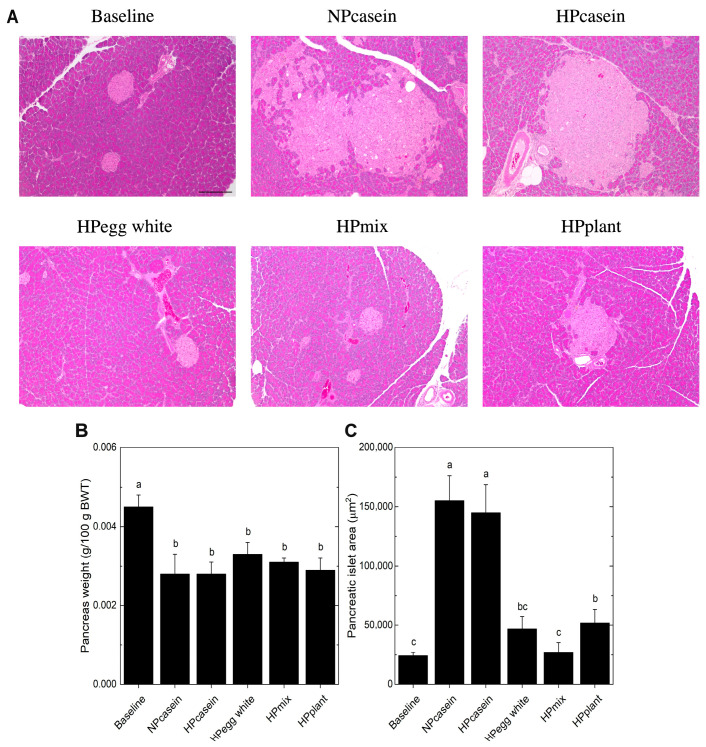
Pancreatic morphology. (**A**) Representative images of hematoxylin and eosin stained pancreas, (**B**) pancreas weight relative to body weight, and (**C**) pancreatic islet size. The pancreatic islets in (**A**) are the lighter pink stained structures. The scale bar shown in (**A**) for the Baseline group represents 200 μm and applies to all panels. Data are mean ± SEM (n = 8 per group for (**B**); n = 5 per group for (**C**), except n = 4 for NPcasein). Columns (**B**,**C**) with different letters are significantly different (*p* < 0.05) from each other.

**Table 1 nutrients-17-03225-t001:** Diet Formulations.

Ingredients (g/kg) ^1^	NP-Casein	HP-Casein	HP-Egg White	HP-Mix	HP-Plant
Cornstarch	404	194	174	205	213
Casein (87.6% protein) ^2^	165	374	0	0	0
Egg white (82.1% protein) ^2^	0	0	398	190	0
Soy protein (78.1% protein) ^2^	0	0	0	95	188
Pea protein (78.6% protein) ^2,3^	0	0	0	95	188
Maltodextrin	132	132	132	132	132
Sucrose	100	100	100	100	100
Cellulose	50	50	50	50	50
AIN-93G mineral mix	35	35	35	35	35
AIN-93VX vitamin mix	10	10	10	10	10
L-cystine	3	6	0	0	0
Choline bitartrate	3	3	3	3	3
Soybean oil ^4^	98	96	98	85	81

^1^ Ingredients from Dyets (Bethlehem, PA, USA) unless otherwise indicated. ^2^ Protein, lipid and carbohydrate content determined by proximate analysis (Bureau Veritas, Mississauga, ON, Canada). ^3^ Now Foods Pea Protein, Non-GMO Vegan, Unflavoured; Manufacturer: Puresource, Guelph, ON, Canada; Sold by: Amazon.com.ca Inc (Vancouver, BC, Canada). ^4^ Soybean oil with 0.02% tert-butylhydroquinone from Dyets (Bethlehem, PA, USA). Abbreviations: AIN-93G, American Institute of Nutrition-93 Growth; NP, normal protein (15% kcal from protein, same as AIN-93G diet); HP, high-protein (35% kcal from protein).

## Data Availability

Data are available upon request from the authors.

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
