# Peer review of "Protein Source Determines the Effectiveness of High-Protein Diets in Improving Adipose Tissue Function and Insulin Resistance in fa/fa Zucker Rats"

_nutrients, 2025, doi:10.3390/nu17203225_

Round 1

Reviewer 1 Report

Comments and Suggestions for Authors

This manuscript:Protein source determines the effectiveness of high-protein diets in improving adipose tissue function and insulin resistance in fa/fa Zucker rats presents a well-designed study comparing the effects of different high-protein diets on adipose tissue function and insulin resistance in fa/fa Zucker rats. The data are supportive of the authors’ conclusions, and the references cited are current and appropriate. 

Several issues should be addressed before the manuscript can be considered for publication:

  1. Sex of animals used – The study utilized only male rats. Since males and females may respond differently due to hormonal influences, the authors should provide justification for the use of only male rats and discuss potential sex-related differences in the Discussion section.

  2. Water intake monitoring – In the Materials and Methods section, it is unclear whether water intake was monitored. Given that water consumption can also influence dietary effects, the authors should clarify whether this variable was controlled or measured.

  3. Figures 2A and 5A – These figures would benefit from clearer presentation. Please include magnification information for each image and add arrows or markers to highlight the observed changes in adipose cells.

  4. Statistical notation – The figures currently use different letters to indicate significant differences. This system may be confusing for readers. It is recommended to use symbols (e.g., *, #, †) to denote statistical significance and clearly indicate in the legend which groups are being compared.

  5. Formatting – In Lines 41 and 63, the references are missing full stops at the end of the sentences. Please correct this punctuation issue.

With these revisions, the manuscript will be much clearer and stronger.

Author Response

1- Sex of animals used– The study utilized only male rats. Since males and females may respond differently due to hormonal influences, the authors should provide justification for the use of only male rats and discuss potential sex-related differences in the Discussion section.

Response:  We appreciate the reviewer’s comment and agree that sex-specific hormonal influences could lead to different responses. We have highlighted this as a limitation in the Discussion (lines 402-407). Our study budget was not sufficient to purchase both male and female fa/fa Zucker rats and which would double the number of analyses.

2- Water intake monitoring– In the Materials and Methods section, it is unclear whether water intake was monitored. Given that water consumption can also influence dietary effects, the authors should clarify whether this variable was controlled or measured.

Response:  Thank you for this observation. Water intake was neither measured nor controlled, as the rats had free access to water. A sentence in the Methods has been revised accordingly (line 105, section 2.1) and we have acknowledged this as a limitation in the Discussion (lines 402-403).

3- Figures 2A and 5A– These figures would benefit from clearer presentation. Please include magnification information for each image and add arrows or markers to highlight the observed changes in adipose cells.

Response:  We would like to clarify that the magnification information is provided by the scale bar, which is shown in the bottom right corner of the Baseline panel in Figure 2A and 5A. We have updated the respective Figure legends (lines 242-243 and 314) to state that “The scale bar shown in (A) for the Baseline group represents 20 μm and applies to all panels.” for Fig 2, and The scale bar shown in (A) for the Baseline group represents 200 μm and applies to all panels.” for Fig 5. To address the request for markers highlighting the differences in these figure panels, we have added material to the legend of Fig 5 that clarifies how the islets appear within the stained section (lines 313-314). This may not have been obvious previously, and given this information there should be no need for other markings on the panel. In Fig 2, the adipocytes are very clear in the figure and it was felt any addition would obscure the general appearance and relative cell size between the treatments.

4- Statistical notation– The figures currently use different letters to indicate significant differences. This system may be confusing for readers. It is recommended to use symbols (e.g., *, #, †) to denote statistical significance and clearly indicate in the legend which groups are being compared.

Response:  We appreciate the reviewer’s comment. The letter-based system is used by many nutrition journals, including Nutrients, and this system allows the statistical comparisons among all groups to be shown as per the post-hoc testing with Duncan’s multiple range test. 

5- Formatting– In Lines 41 and 63, the references are missing full stops at the end of the sentences. Please correct this punctuation issue.

Response:  Thank you for pointing out these errors. They have been corrected.

Reviewer 2 Report

Comments and Suggestions for Authors

The aim off this study was to investigate if the source of protein (animal or plant) has an impact on adipose tissue function and insulin sensitivity. The conclusion is that the source of the protein in high protein diets significantly influences metabolic outcomes in obese rats. One issue that was not reported was the energy intake. Figure 1 would suggest that the rats on the Plant derived high protein diets ate more than those on the Animal derived high protein diets.  Strangely, the Plant derived high protein diet rats put on more weight and more fat tissue but they had smaller fat cell sizes and more adiponectin.  The other interesting finding was that of the two animal derived proteins, egg white protein had very different response to casein.

Author Response

1- The aim off this study was to investigate if the source of protein (animal or plant) has an impact on adipose tissue function and insulin sensitivity. The conclusion is that the source of the protein in high protein diets significantly influences metabolic outcomes in obese rats. One issue that was not reported was the energy intake. Figure 1 would suggest that the rats on the Plant derived high protein diets ate more than those on the Animal derived high protein diets.  Strangely, the Plant derived high protein diet rats put on more weight and more fat tissue but they had smaller fat cell sizes and more adiponectin.  The other interesting finding was that of the two animal derived proteins, egg white protein had very different response to casein.

Response:  We appreciate the reviewer’s observations regarding the responses to the plant-derived high-protein diet. The limitation regarding energy intake has been acknowledged and discussed in the new Discussion paragraph on limitations (lines 402-403 and 407-416).

Reviewer 3 Report

Comments and Suggestions for Authors

In conclusion: 

This article is important because it offers new avenues for intervention in nutrient-related metabolic diseases that regulate obesity, diabetes, and complications.

Since common mechanisms are likely present in animals, it is important that these findings be reevaluated in humans, starting with mice genetically predisposed to diabetes.

I have no errors or corrections to report to the authors.

Author Response

In conclusion: 

This article is important because it offers new avenues for intervention in nutrient-related metabolic diseases that regulate obesity, diabetes, and complications.

Since common mechanisms are likely present in animals, it is important that these findings be reevaluated in humans, starting with mice genetically predisposed to diabetes.

I have no errors or corrections to report to the authors.

Response:  We thank the reviewer for their positive comments.